# WMAdapter: Adding WaterMark Control to Latent Diffusion Models

**Hai Ci**[1]  **Yiren Song**[1]  **Pei Yang**[1]  **Jinheng Xie**[1]  **Mike Zheng Shou**[1]

## Abstract

Watermarking is essential for protecting the copyright of AI-generated images. We propose WMAdapter, a diffusion model watermark plugin that embeds user-specified watermark information seamlessly during the diffusion generation process. Unlike previous methods that modify diffusion modules to incorporate watermarks, WMAdapter is designed to keep all diffusion components intact, resulting in sharp, artifact-free images. To achieve this, we introduce two key innovations: (1) We develop a contextual adapter that conditions on the content of the cover image to generate adaptive watermark embeddings. (2) We implement an additional finetuning step and a hybrid finetuning strategy that suppresses noticeable artifacts while preserving the integrity of the diffusion components. Empirical results show that WMAdapter provides strong flexibility, superior image quality, and competitive watermark robustness. Code: `https://github.com/showlab/WMAdapter`

## 1. Introduction

With the widespread adoption of diffusion models (Ho et al., 2020; Podell et al., 2023; Song et al., 2020; Rombach et al., 2022; Ci et al., 2023; Zhang et al., 2023a; Wang et al., 2024), diffusion-generated images are proliferating across media and the internet. While these models meet the demand for high-quality creative content, their misuse raises significant concerns about copyright protection and the security of images against deepfakes (Westerlund, 2019; Song et al., 2024b;a). Watermarking technology (Cox et al., 2007) provides a tailored solution for resolving copyright disputes and identifying the sources of forgeries.

Previous watermarking methods added watermarks to im-

ages in a post-hoc way through frequency domain transformations (Cox et al., 2007; Lin et al., 2001; Xia et al., 1998) or encoder-decoder networks (Zhu et al., 2018; Tancik et al., 2020; Zhang et al., 2019). However, in the context of watermarking diffusion images, post-hoc methods introduce additional workflows and unable to fully leverage the rich latent space provided by the image generation process. Recently, more efforts (Zhao et al., 2023b; Fernandez et al., 2023; Min et al., 2024; Xiong et al., 2023; Lei et al., 2024; Meng et al., 2024; Yang et al., 2024b; Ci et al., 2024) have focused on leveraging the characteristics of the diffusion process to seamlessly integrate watermarking into the diffusion pipeline, known as diffusion-native watermarking. Among these, Stable Signature (Fernandez et al., 2023) proposed a method that fine-tunes the VAE decoder of a latent diffusion model (Rombach et al., 2022) using a pretrained watermark decoder (Zhu et al., 2018). This approach has shown promising results. However, it requires fine-tuning a separate VAE decoder for each unique watermark, making it difficult to scale to millions of keys as required in large-scale commercial scenarios where each user may need a unique key. Additionally, the tuning of VAE decoder on a small amount of data results in blurry and lens flare-like artifacts (see Fig. 7).

Recent works (Bui et al., 2023; Xiong et al., 2023; Min et al., 2024; Meng et al., 2024; Zhang et al., 2024; Kim et al., 2023; Nguyen et al., 2023) have explored watermark plugins for diffusion models. These plugins accept arbitrary watermark keys and generate watermark embeddings without requiring per-watermark finetuning, thereby addressing the scalability issue. However, these methods typically generate watermark embeddings without considering the image content (Kim et al., 2023; Xiong et al., 2023; Bui et al., 2023) (i.e., they are context-less) and often require finetuning or modifying diffusion modules to incorporate the watermark embeddings (Kim et al., 2023; Xiong et al., 2023; Feng et al., 2024) . Tab. 1 compares several watermarking methods. Unfortunately, finetuning the original diffusion pipeline or making intrusive modifications often leads to a significant drop in image quality, resulting in blurriness or noticeable artifacts. Fig. 1 illustrates the image quality of different methods, where artifacts introduced by other methods are evident. Find more examples in Fig. 12.

We propose an innovative watermark plugin solution —

---

[1]Show Lab, National University of Singapore, Singapore. Correspondence to: Mike Zheng Shou <mike.zheng.shou@gmail.com>.

*Proceedings of the 42$^{nd}$ International Conference on Machine Learning*, Vancouver, Canada. PMLR 267, 2025. Copyright 2025 by the author(s).

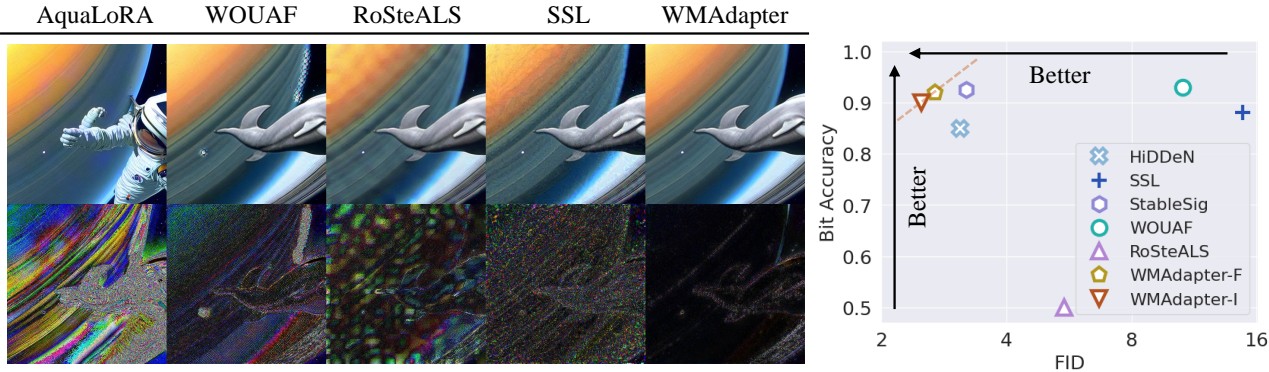

Figure 1: WMAdapter introduces minimal artifacts, providing better accuracy-quality tradeoff.

Table 1: Comparison of several diffusion watermarking methods. They all tend to introduce noticeable artifacts or produce blurry images.

| | Modified Diffusion Modules | Scalable | Imperceptible |
|---|---|---|---|
| AquaLoRA (Feng et al., 2024) | UNet Backbone | ✓ | ✗ |
| StableSig (Fernandez et al., 2023) | VAE Decoder | ✗ | ✗ |
| WOUAF (Kim et al., 2023) | VAE Decoder | ✓ | ✗ |
| RoSteALS (Bui et al., 2023) | No | ✓ | ✗ |
| Ours | No | ✓ | ✓ |

WMAdapter (Fig. 2). Its core design philosophy focuses on preserving the integrity of the original diffusion pipeline to produce high-quality images. We do not modify any parameters of the pretrained diffusion modules. So how do we conceal the watermark information and ensure its robustness? We introduce two key innovations: (1) We propose a novel **Contextual Adapter** structure that conditions on the cover image features to generate content-aware watermark embeddings (hence "contextual"). Intuitively, this allows the adapter to better identify areas of the image that are more suitable for hiding the watermark, enhancing concealment and robustness. To fully leverage diffusion features while reducing computational overhead, our Contextual Adapter extracts image features from the intermediate layers of the diffusion VAE decoder. Unlike ControlNet plugins (Zhang et al., 2023b; Min et al., 2024), which use a heavy UNet structure (Ronneberger et al., 2015), the Contextual Adapter is lightweight, totaling only 1.3MB in parameters, and enables watermarking an image in just 30ms. (2) We introduce an additional finetuning stage with a novel **Hybrid Finetuning** strategy to further enhance image quality. To preserve the original diffusion modules, our Hybrid Finetuning strategy involves jointly finetuning the adapter and the diffusion VAE decoder during training for alignment, and then using the original VAE decoder during inference. This approach effectively suppresses noticeable artifacts and significantly improves image sharpness. We summarize our contributions

as follows:

1. We introduce **WMAdapter**, a novel diffusion watermarking solution with an innovative design philosophy. It embeds watermarks non-intrusively during the diffusion process, thereby preserving the integrity of the diffusion pipeline and producing high-quality images.

2. Methodologically, we propose **Contextual Adapter** and **Hybrid Finetuning** to achieve non-intrusive watermarking, ensuring both watermark robustness and generation quality.

3. Experimental results demonstrate that WMAdapter effectively suppresses noticeable artifacts and offers better accuracy-quality tradeoffs compared to prior post-hoc and diffusion-native watermarking methods.

## 2. Related Work

### 2.1. Post-hoc Watermarking

Post-hoc methods include traditional frequency domain transformation methods (Cox et al., 2007), optimization-based methods (Fernandez et al., 2022b; Kishore et al., 2021), and encoder-decoder methods (Zhu et al., 2018; Tancik et al., 2020; Jia et al., 2021; Sander et al., 2024). Different methods have different aims. For instance, Kishore et al. (2021) emphasizes hiding more bits, Zhu et al. (2018)

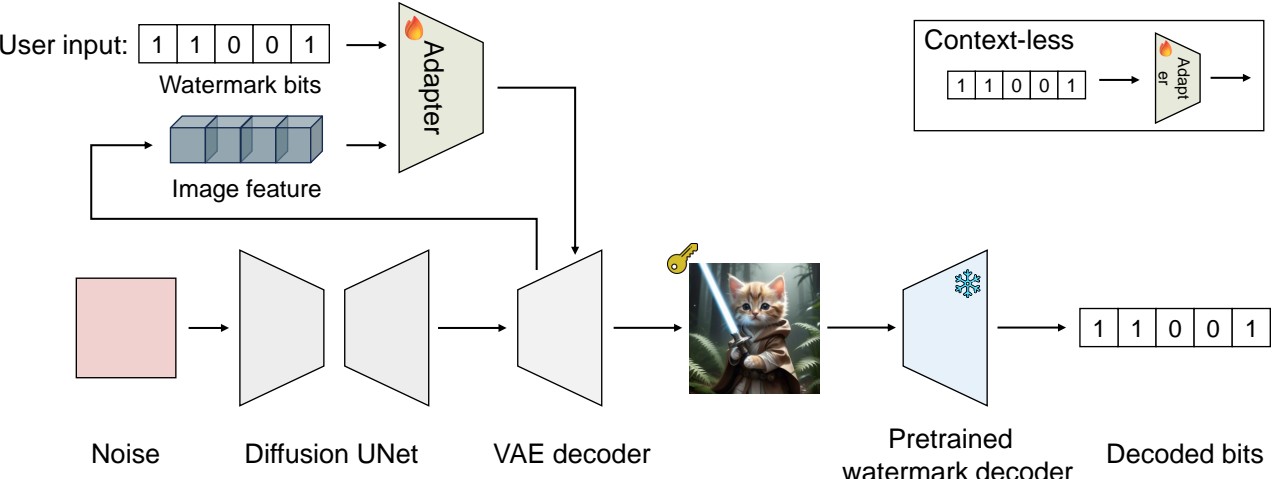

Figure 2: Framework overview. WMAdapter is plugged onto the VAE decoder. It takes user input watermark bits and image features from the VAE decoder, imprinting the watermark on-the-fly during VAE decoding. In contrast, traditional context-less adapters take only watermark conditions as input. The image and icons credit to (Freepik-Flaticon, 2024).

and Jia et al. (2021) prioritizes robustness against JPEG compression.

## 2.2. Diffusion Native Watermarking

According to the location of the watermark, we classify diffusion-native watermarking methods into two categories. **Adding to initial noise:** Tree-Ring (Wen et al., 2023) adds watermarks to the frequency of initial noise, achieving remarkable robustness. Subsequent methods (Yang et al., 2024b; Ci et al., 2024; Lei et al., 2024) improves its multi-key identification capabilities. However, these methods significantly alter the layout of the generated images, which is not desirable in some production scenarios. **Adding to latent space:** Other methods leverage the latent space of the VAE (Bui et al., 2023; Meng et al., 2024; Zhang et al., 2024; Xiong et al., 2023; Kim et al., 2023; Fernandez et al., 2023) or diffusion backbone (Feng et al., 2024). However, they either generate content-agnostic watermark embeddings or modify the original diffusion modules, often resulting in lower image quality. In contrast, WMAdapter prioritizes image quality through novel contextual designs while preserving the integrity of the entire diffusion pipeline. Stable Messenger (Nguyen et al., 2023) is a recent method that also generates content-aware watermarks. However, they mainly focus on improving message accuracy and their model design is different from ours.

## 3. Method

In this section, we will introduce the framework of WMAdapter, detail its contextual structure, and discuss the training and fine-tuning strategies.

### 3.1. Framework Overview

Fig.2 illustrates the overall framework of WMAdapter. WMAdapter is a plug-and-play watermark module that can be directly attached to the VAE decoder of a latent diffusion model (Rombach et al., 2022). It imprints the watermark during image generation, seamlessly integrating into the diffusion generation workflow. WMAdapter employs a novel contextual adapter structure, which takes both watermark bits and image features from the VAE decoder as input and outputs feature residuals containing watermark information. Watermarked images can be directly fed into a pretrained watermark decoder, such as HiDDeN (Zhu et al., 2018), to retrieve the watermark information.

The training of WMAdapter consists of two stages: large-scale training and fast finetuning. In the training stage, we freeze the VAE decoder and the watermark decoder and train only the Adapter on a large scale dataset. We then finetune the Adapter and VAE decoder on a small amount of data. Specifically, we present a novel hybrid finetuning strategy that is able to suppress tiny artifacts and significantly enhance generation quality. We also discuss several different strategies concerning different tradeoffs between robustness and quality.

### 3.2. Contextual Adapters

In this section, we provide a detailed overview of the contextual structure of WMAdapter. Fig. 3 (*Left*) illustrates the internal structure of WMAdapter, which comprises a series of independent *Fuser* modules. Each *Fuser* $\phi_i(\cdot)$ is attached before a corresponding VAE decoder block $i$. It receives both VAE feature $f_i$ and watermark bits $w$ as inputs, and

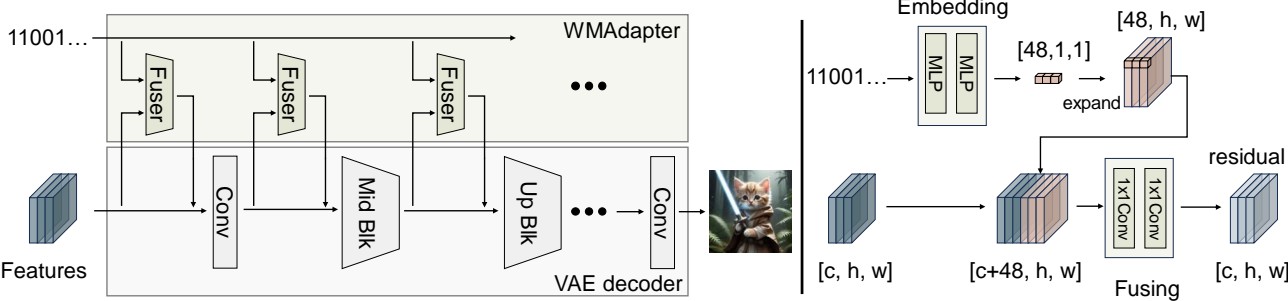

Figure 3: The architecture of WMAdapter. *Left:* The structure of WMAdapter. It comprises several independent Fusers with identical structures. *Right:* The structure of Fuser. It consists of a watermark Embedding module and a Fusing module.

outputs a feature residual $y_i$ to update $f_i$. Formally,

$$y_i = \phi_i (f_i, w),$$
$$f'_i = f_i + y_i. \quad (1)$$

We put a total of 6 *Fusers* before the Conv Block, Middle Block and four Up Blocks in the kl-f8 VAE decoder used by Stable Diffusion (Rombach et al., 2022).

Fig. 3 (*Right*) illustrates the internal structure of an Fuser. An Fuser consists of two main components: the Embedding module and the Fusing module. The Embedding module maps the 01 bit sequence into a 48-dimensional watermark feature vector. This feature vector is then expanded along the width and height dimensions to produce a watermark feature map with the same dimensions as the image feature. The image feature and watermark feature are concatenated along the channel dimension and fed into the Fusing module, which outputs the image feature residuals. Keeping lightweight in mind, we use two MLPs with 256 intermediate feature channels for the Embedding module, and two 1x1 convolutions with half the image feature channels $\frac{c}{2}$ as intermediates for the Fusing module. We employ LeakyReLU as the non-linearity. The total parameters of WMAdapter are only 1.3M, making it a small and efficient plugin.

### 3.3. Training

In the training stage, we use a pretrained watermark decoder to decode watermark bits from the watermarked images. We freeze the watermark decoder and the VAE decoder, and only train the Adapter. Why do we use a pretrained decoder instead of training a watermark decoder from scratch along with the Adapter? We observe that training an encoder/decoder pair from scratch, as post-hoc methods do, typically requires significant training effort. For example, HiDDeN takes 300 epochs to converge on the COCO dataset. The situation gets worse when trained with a diffusion pipeline. WOUAF (Kim et al., 2023) takes about 10 days. Using a pretrained post-hoc decoder facilitates efficient knowledge transfer, allowing WMAdapter to

converge in just 1-2 epochs. Note that this will not bring serious security risks, because there are hundreds of different open-source decoders. We use two types of losses as our objective: the consistency loss between the watermarked image $x_w$ and the unwatermarked image $x$, and the accuracy of decoded bits. The total loss function is defined as:

$$\mathcal{L} = \lambda_1 \mathcal{L}_{mae}(x, x_w) + \lambda_2 \mathcal{L}_{lpips}(x, x_w)$$
$$+ \lambda_3 \mathcal{L}_{vgg}(x, x_w) + \lambda_4 \mathcal{L}_{bce}(w, w') \quad (2)$$

where the first three terms represent image consistency losses. We use MAE and LPIPS loss (Zhang et al., 2018) to maintain consistency with VAE pretraining (Rombach et al., 2022). Additionally, we include a Watson-VGG loss (Czolbe et al., 2020) similar to Stable Signature (Fernandez et al., 2023) to enhance human visual preference. For watermark decoding accuracy, we use binary cross-entropy loss bewteen decoded bits $w'$ and input bits $w$. We empirically set $\lambda_1, \lambda_2, \lambda_3, \lambda_4$ to 0.2, 0.2, 0.08, 1.0, respectively.

### 3.4. Hybrid Finetuning

After the training stage, we obtain a watermark adapter that performs well in both accuracy and image quality (Sec. 4.4.2). However, when we zoom in on the generated images, grid-like artifacts can sometimes be observed (Fig. 6). To further improve image quality and eliminate these tiny artifacts, we introduce a fine-tuning stage on a small amount of data. On top of the first stage training losses, we incorporate an additional total variation loss (et al, 2024) on the watermarked images to enhance smoothness, setting its weight to 0.02.

Further, we present a novel Hybrid Finetuning strategy. Concretely, we finetune both the Adapter and the VAE decoder, but use the fine-tuned Adapter and the original VAE decoder for inference. Fig. 4 distinguishes this strategy from two other classic finetuning strategies: Fixed and Joint Finetuning. The Fixed Finetuning strategy uses the same training

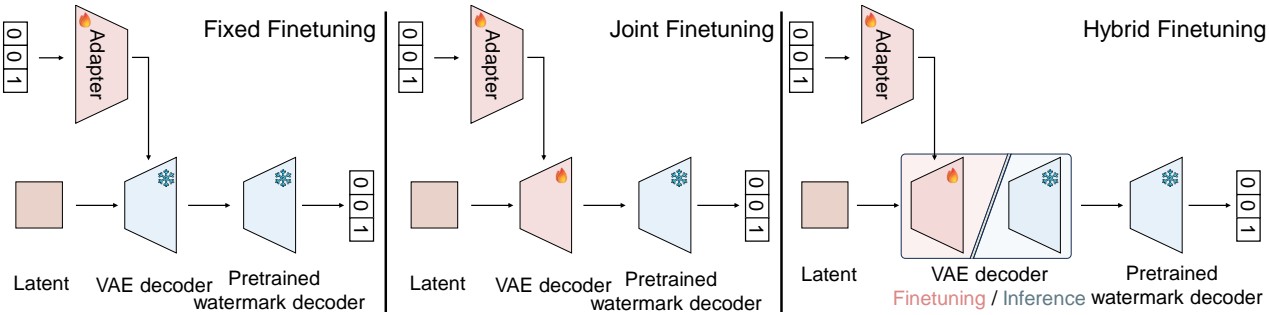

Figure 4: Illustration of 3 different finetunig strategies. They differ in how to treat the VAE decoder.

approach as in the first stage, fixing the VAE decoder and quickly finetuning the Adapter with a high learning rate. The Joint Finetuning strategy jointly finetunes the Adapter and the VAE decoder, using both finetuned copies for inference.

Sec. 4.4.2 will give a side-by-side comparison between these three finetuning strategies. In short, Hybrid Finetuning can effectively suppress noticeable artifacts and, by keeping the VAE intact, produces the sharpest and clearest images while maintaining the plug-and-play advantage, making it ideal for commercial image generation products which require high image quality.

### 3.5. Discussion

WMAdapter is designed with a strong emphasis on image quality, particularly in suppressing noticeable artifacts in generated images. We introduce the **Contextual Adapter** and the **Hybrid Finetuning**, non-intrusive watermarking methods that achieve this goal by preserving the integrity of the diffusion pipeline. This fundamentally distinguishes our approach from other diffusion watermarking methods that embed watermarks at the expense of image quality and introduce noticeable artifacts. We want to highlight the importance of high-quality, artifact-free watermarked images for generative products, as no user wants to receive images with visible flaws. The Experiment Section demonstrates that our method successfully combines scalability, high-quality image generation, and watermark robustness.

## 4. Experiments

### 4.1. Experimental Setup

**Model and dataset** We experiment with a popular latent diffusion model Stable Diffusion 2.1 (Rombach et al., 2022) and its associated kl-f8 VAE. We adopt the pretrained watermark decoder from HiDDeN (Zhu et al., 2018). The checkpoint we use was pretrained by (Fernandez et al., 2023), encoding 48-bits watermark information. This checkpoint is also used to finetune Stable Signature (Fernandez et al., 2023). Thus, our adapter can be directly compared with (Fer-

nandez et al., 2023). ALL training and finetuning steps are performed on MS-COCO 2017 (Lin et al., 2014) training set. Validation is performed on COCO 2017 validation set. We train and evaluate our adapters on images at resolution $512 \times 512$. For images smaller than this size, we resize their shorter edge to 512, then center crop to get a $512 \times 512$ image.

**Training strategies** For the first stage training, we adopt 8 $\times$ NVIDIA A5000 GPUs of 24 GB memory, with per-GPU batchsize of 2, AdamW optimizer (Loshchilov & Hutter, 2017), a learning rate of 5e-4. We train the model for 2 epochs, taking about 5 hours. For the second stage finetuning, we use a single A5000 GPU. We set the mini-batch to 2. We also use the AdamW optimizer and a start learning rate of 5e-4. However, we adopt a per-step cosine learning rate decay with 20 warm-up steps. Unless otherwise specified, the total fine-tuning process defaults to 2,000 steps, lasting for about 50 minutes. Different finetuning strategies result in several different adapter variants. We use Adapter-*B*, Adapter-*F*, Adapter-*V*, and Adapter-*I* to denote the adapters obtained by No Finetuning, Fixed Finetuning, Joint Finetuning and Hybrid Finetuning, respectively.

**Evaluation metric** Following previous conventions (Zhu et al., 2018; Fernandez et al., 2022b; 2023), we use average bit accuracy to evaluate the watermarking performance of our adapter. Bit accuracy is defined by the ratio of correctly decoded bits in a 48-bit watermark sequence. Apart from the bit accuracy, we also report the tracing accuracy among different numbers of users following concurrent works (Min et al., 2024; Ci et al., 2024). We adopt the evaluation protocol of (Min et al., 2024). Concretely, we construct user pools of different sizes, ranging from $10^4$ to $10^6$, to evaluate the accuracy of user tracing at different scales. Each user is assigned a unique key. For each user pool. we randomly select 1,000 users and watermark 5 images per user, resulting in 5,000 watermarked images. For each of the 5,000 images, we find the best match among the user pool and check if it's a correct match. Tracing accuracy is then averaged over all 5,000 images. To evaluate the detection performance,

we report TPR@FPR$10^{-6}$. Concretely, we assume the bits decoded from the natural images following Bernoulli distribution with parameter 0.5. Then the number of matched bits $M$ follows a binomial distribution with parameters $(48, 0.5)$. So we have the false detection rate as a function of threshold $\tau$: $FPR(\tau) = \mathcal{P}(M > \tau) = \mathcal{I}_{0.5}(\tau + 1, 48 - \tau)$, where $\mathcal{I}$ is the incomplete beta function. We control $FPR = 10^{-6}$ and calculate the corresponding $\tau$, then we evaluate TPR with this threshold.

In addition to accuracy measurements, we are also interested in the watermark's invisibility and image generation quality. We report the Peak Signal-Noise-Ratio (PSNR) between images before and after watermarking and Fréchet Inception Distance (FID) (Heusel et al., 2017) between watermarked images and images from coco val set. Typically, higher PSNR leading to sharper and clearer images. While lower FID means the watermarked images have higher fidelity and more closely resemble the real images in terms of appearance and variety.

## 4.2. Comparison With Other Methods

**Accuracy and image quality** We compare our method with three post-hoc watermarking methods SSL (Fernandez et al., 2022b), StegaStamp (Tancik et al., 2020), and HiDDeN (Zhu et al., 2018). SSL bases on iterative optimization to get the watermark, while StegaStamp and HiDDeN are encoder-decoder based methods. For HiDDeN, we use the model provided by (Fernandez et al., 2023), which is enhanced with a JND mask (Fernandez et al., 2022a) for better image quality. We also compare with three recent diffusion-native watermarking methods RoSteALS (Bui et al., 2023), WOUAF (Kim et al., 2023) and Stable Signature (Fernandez et al., 2023). Note that all these methods do not alter the image layout during watermarking.

As shown in Tab. 2, WMAdapter-*I* achieves the best image quality among all methods, excelling in both PSNR and FID. Its PSNR and FID improve over the baseline, Stable Signature, by approximately 17% and 22%, respectively. In contrast, Stable Signature produces blurrier images with lens flare artifacts (Sec. 4.5) due to fine-tuning of the VAE decoder, resulting in lower PSNR and FID scores. WMAdapter-*I* shows even greater improvements compared to SSL (5% and 83%), RoSteALS (14% and 55%), and WOUAF (38% and 81%), as these methods introduce larger artifacts greatly degrading quality metrics (See Fig. 12 for artifacts).

In terms of watermark detection performance, our methods achieve perfect TPR, outperforming HiDDeN, WOUAF, and Stable Signature. For bit accuracy, while SSL excels in single attack scenario, it is more sensitive to combined attacks. Both WMAdapter-*F* and WMAdapter-*I* surpass SSL, HiDDeN and RoSteALS under combined attacks, trailing the top-performing methods by only 0.01 and 0.03, respectively, while still maintaining competitive robustness. Overall, WMAdapter achieves a *better robustness-quality tradeoff*, which can be seen in Fig. 1 (*right*).

**Tracing accuracy** Since certain watermarking methods, such as Wen et al. (2023), don't incorporate the concept of bits or use tracing accuracy as an alternative evaluation protocol (Min et al., 2024), we further compare the tracing accuracy in Tab. 3. We can see that our adapters achieve nearly perfect tracing accuracy with different scales of users. Tree-Ring (Wen et al., 2023) achieves zero tracing accuracy due to its design flaws uncovered by Ci et al. (2024). WAD-IFF (Min et al., 2024) is a concurrent effort, which employs HiDDeN decoder to finetune a UNet watermark plugin for diffusion models. We can see that its tracing accuracy gradually drops as the scale grows despite they employ a heavier adapter (∼900MB params). Both ours and Stable Signature perform consistently at different user scales. Notably, Stable Signature has higher average bit accuracy but gets slightly worse tracing accuracy than ours. We attribute this to its larger performance variance among different keys.

**Summary** Unlike other methods with significant drawbacks—such as RoSteALS, SSL, and WOUAF, which introduce noticeable artifacts and result in significantly lower FID scores, or StableSignature, which lacks scalability—our approach delivers high image quality, scalability, and competitive accuracy simultaneously. In all three aspects, WMAdapter-*I* consistently outperforms HiDDeN, providing a better overall tradeoff.

## 4.3. Robustness to More Attacks

**Other transformations and intensities** Fig. 8 evaluates against more image transformations and intensities. Our adapters achieve comparable performance to the baseline Stable Signature under various levels of attacks, while offering flexibility, scalability and higher image quality.

**Regeneration attack** Recent work (Zhao et al., 2023a; Liu et al., 2024) has demonstrated the potential of regeneration attacks in watermark removal. We evaluate the robustness of WMAdapter against three different regeneration methods introduced in Zhao et al. (2023a): one diffusion-based (Zhao et al., 2023a) and two VAE-based methods (Ballé et al., 2018; Cheng et al., 2020). For Ballé et al. (2018); Cheng et al. (2020), we assess performance at compression rates of 1-6 and 1-8, respectively. Fig. 5 presents the Accuracy-PSNR curve. We observe that the three regeneration attacks require a PSNR drop of 4-6 dB to successfully remove our watermark. In contrast, only a 2 dB reduction in image quality is needed to remove the watermark of Stable Signature. This demonstrates that our

Table 2: Comparison with other watermarking methods on generation quality and robustness. All methods are evaluated on COCO 2017 val set (Lin et al., 2014) with image size $512 \times 512$. Since Stable Signature (Fernandez et al., 2023) requires finetuning of separate VAE decoders to embed different keys, we report its average results on 10 randomly sampled keys. We report TPR@FPR$10^{-6}$ for detection performance. For robustness, we use Crop 0.3, JPEG 80, Brightness 1.5.

| | Method | PSNR ↑ | FID ↓ | TPR ↑ | Bit Accuracy ↑ | | | | |
| | | | | | None | JPEG | Crop | Bright | Comb |
|---|---|---|---|---|---|---|---|---|---|
| *Post* | SSL | 33.0 | 14.8 | **1.00** | **1.00** | **0.99** | 0.97 | 0.98 | 0.88 |
| | HiDDeN | 34.1 | 3.1 | 0.99 | 0.98 | 0.84 | 0.97 | 0.98 | 0.85 |
| | StegaStamp | 29.3 | 9.9 | **1.00** | 0.96 | 0.96 | 0.49 | 0.94 | 0.49 |
| *Native* | RoSteALS | 30.4 | 5.5 | **1.00** | 0.99 | **0.99** | 0.50 | 0.96 | 0.50 |
| | WOUAF | 25.3 | 13.5 | 0.97 | 0.99 | **0.99** | 0.94 | 0.97 | **0.93** |
| | Stable Signature | 29.7 | 3.2 | 0.99 | 0.99 | 0.93 | **0.99** | **0.99** | **0.93** |
| | WMAdapter-*F* | 33.1 | 2.7 | **1.00** | 0.99 | 0.92 | **0.99** | **0.99** | 0.92 |
| | WMAdapter-*I* | **34.8** | **2.5** | **1.00** | 0.98 | 0.90 | 0.97 | 0.97 | 0.90 |

Table 3: Accuracy of tracing different numbers of keys. All methods are evaluated on COCO dataset (Lin et al., 2014). For WADIFF* (Min et al., 2024), the number is reported by its original paper.

| Method | Trace $10^4$ | Trace $10^5$ | Trace $10^6$ |
|---|---|---|---|
| WADIFF* | 0.982 | 0.968 | 0.934 |
| Tree-Ring | 0.000 | 0.000 | 0.000 |
| Stable Signature | 0.999 | 0.999 | 0.998 |
| WMAdapter-*F* | **1.000** | **1.000** | **1.000** |
| WMAdapter-*I* | **1.000** | 0.999 | 0.999 |

method exhibits better robustness against regeneration attacks.

**Adversarial attack** Adversarial attack relies on PGD (Madry, 2017) optimization to generate adversarial noise targeting the watermark decoder. Based on access to the watermark decoder, these attacks are categorized as white-box and black-box. In black-box settings, a binary classifier is trained to identify watermarked images, and adversarial noise is then optimized to mislead this classifier, disrupting the watermark. This is commonly referred to as a surrogate detector attack (Saberi et al., 2023; Jiang et al., 2023; An et al., 2024; Lukas et al., 2023). We follow the implementation of An et al. (2024) and demonstrate our method's robustness against both white-box (An'24-wb △) and black-box attacks (An'24 ▽) in Fig. 5. Notably, both WMAdapter and Stable Signature exhibit strong robustness against black-box adversarial attacks, with a bit accuracy drop of about 0.02 and TPR drop less than 0.01. In white-box scenarios, where attackers have full access to the watermark decoders, the watermarks can be easily disrupted with minimal impact on image quality.

**Query-based attack** Another common black-box attack is the query-based attack, which defines a blending process that transitions from a random image to a given watermarked image. During this process, it repeatedly queries the watermark decoder API to determine whether the current blended image contains a watermark, aiming to identify the image with the minimal perturbation that successfully removes the watermark. We adopt the WEvade-B-Q approach from Jiang et al. (2023) and set the detection threshold $\tau$ to control $FPR = 10^{-6}$. Our observations show that the query-based attack can successfully evade watermark detection for both WMAdapter and Stable Signature, achieving a success rate of 1.0 (i.e., $TPR = 0$). However, this method results in significant image quality degradation, with the final attacked images averaging a PSNR of approximately 8 dB.[1]

**Steganalysis attack** Yang et al. (2024a) propose averaging multiple watermarked images to extract content-agnostic watermark patterns for removal or forgery. However, the contextual adapter in WMAdapter adapts watermark patterns based on image layout, making it naturally robust to this type of attack—achieving no bit accuracy drop on a 5k image averaging evaluation.

### 4.4. Ablation Study

#### 4.4.1. WHY CONTEXTUAL ADAPTER?

Tab. 4 compares different adapter variants after the first stage training. We can find that using the contextual adapter structure is crucial for both watermark accuracy and image quality, improving bit accuracy by 0.02 and PSNR by a significant number of 4.1 db compared with the context-

---

[1]We did not include this method in Fig. 5 because the resulting image quality is far outside the scope of the comparisons shown in the figure.

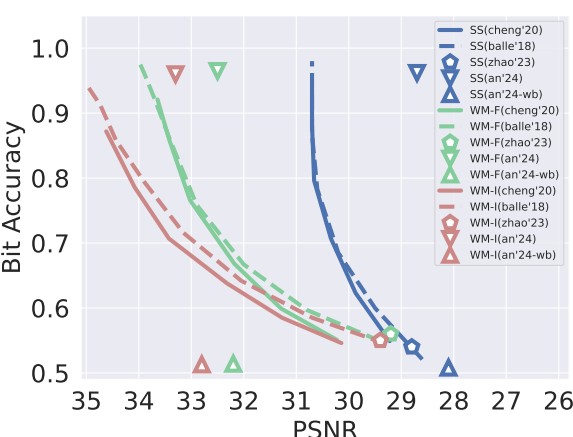

Figure 5: Against various regeneration and adversarial attacks. SS: Stable Signature.

less structure. This result well supports our motivation that the watermark encoder should be aware of the cover image content to generate high quality embedding. Note that SOTA watermarking methods still use the context-less structure to encode watermark (Xiong et al., 2023; Kim et al., 2023; Bui et al., 2023). Contextual adapter provides a simple yet promising approach for further improvement. Another key design is to use 1x1 conv in the adapter, because we found that 3x3 conv suffers from unstable training.

Table 4: Comparison between adapter structures.

|  | Contextual | Context-less | Conv $3 \times 3$ |
|---|---|---|---|
| Bit Acc | **0.99** | 0.97 | 0.49 |
| PSNR | **32.8** | 28.7 | 12.0 |

### 4.4.2. ROLE OF FINETUNING

Tab. 5 and Fig. 6 compare different finetuning strategies quantitatively and qualitatively. From Tab. 5, we can see that Adapter-*B* achieves good numerical results. However, upon closer inspection of the generated images, subtle grid-like artifacts become noticeable. If we freeze the VAE decoder and perform a quick fine-tuning for 2k steps using a large learning rate, resulting in Adapter-*F*. We find that PSNR and SSIM metrics further improve, though the artifacts persisted.

Hybrid Finetuning (Adapter-*I*) further suppresses artifacts. Since the VAE remains unaltered during inference, it produces the sharpest and most visually appealing images, with PSNR improving significantly to 34.8 dB. This improvement comes at the minor cost of a 0.02 decrease in bit accuracy under combined attacks.

Joint Finetuning (Adapter-*V*) significantly degrades all image quality metrics. As shown in Fig. 6, Joint Finetuning

Table 5: Comparison between different finetuning strategies. "Adapter-*B*" means no extra finetuning. Bit Acc is evaluated under combined attacks.

|  | Bit Acc | PSNR | SSIM | FID |
|---|---|---|---|---|
| Adapter-*B* | 0.92 | 32.8 | 0.94 | 2.7 |
| Adapter-*F* | 0.92 | 33.1 | 0.95 | 2.7 |
| Adapter-*I* | 0.90 | **34.8** | **0.96** | **2.5** |
| Adapter-*V* | 0.92 | 29.9 | 0.87 | 3.1 |

results in smoother but blurrier images. It also introduces noticeable lens flare artifacts, which are commonly observed in methods such as Stable Signature (Fernandez et al., 2023), FSW (Xiong et al., 2023), AquaLoRA (Feng et al., 2024), and WOUAF (Kim et al., 2023), as they all modify diffusion components to embed the watermark. This observation supports our core idea that preserving the integrity of the original diffusion pipeline is crucial for high-quality generation.

Considering both numerical results and visual artifacts, Adapter-*F* and Adapter-*I* offer better accuracy-quality trade-offs. Therefore, we adopt these two as our default choices. Note that all adapter variants incorporate an additional total variation loss during the second stage finetuning. While this loss helps produce visually smoother images and provides a 0.1 PSNR improvement, it does not reduce artifacts (Fig. 6). Applying it during the first stage training can lead to overly smoothed images.

### 4.4.3. RESULTS ON DIFFERENT VAES

We train several watermark adapters for VAEs used by SD1.5&2.1 (Rombach et al., 2022), SDXL (Podell et al., 2023) and DiT (Peebles & Xie, 2023) (kl-f8-mse) at resolution $512 \times 512$. We compare the adapters before the finetuning stage. Tab. 6 shows the results. We observe that WMAdapter consistently performs well across various VAEs, making it applicable in a wide range of contexts. The PSNR of SDXL adapter is lower compared to SD2.1 and DiT VAE. This may be caused by the resolution mismatch.

We further evaluate the zero-shot transferability of WMAdapter across different VAEs. Specifically, we directly apply the adapter trained on SD2.1 to the SD1.5 VAE and observe that it effectively handles SD1.5 image latents with minimal performance degradation. This empirical result highlights the zero-shot generalization potential of WMAdapter to various customized Stable Diffusion VAEs.

### 4.5. Qualitative Results

We qualitatively compare WMAdapter with the baseline method, Stable Signature (Fernandez et al., 2023) in Fig. 7.

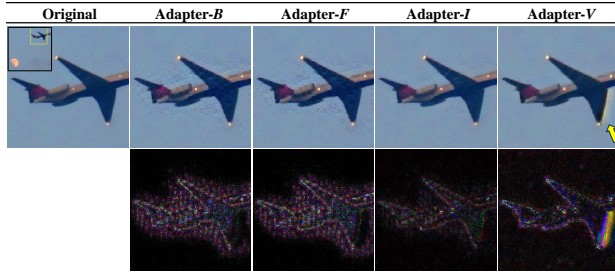

Figure 6: Qualitative comparison between different fine-tuning strategies. Adapter-*B* and Adapter-*F* produces tiny grid-like artifacts. Finetuning with VAE (Adapter-*I* and -*V*) alleviates this issue. Using fintuned VAE at inference time (Adapter-*V*) leads to lens flare artifact. Using original VAE (Adapter-*I*) achieves the most visually appealing results. Zoom in for best view.

Table 6: Evaluation on VAEs from different models.

|         | SD1.5 | SD2.1 | SDXL | DiT  |
| ------- | ----- | ----- | ---- | ---- |
| Bit Acc | 0.99  | 0.99  | 0.99 | 0.99 |
| PSNR    | 32.1  | 32.8  | 31.2 | 32.4 |

We can observe that Stable Signature tends to produce lens flare artifacts, as indicated by the yellow arrows. We attribute this issue to the modification of VAE decoder. In contrast, Adapter-*F* and Adapter-*I* greatly suppress this noticeable artifact by preserving the integrity of all diffusion components. As shown in columns (*C*)(*D*), our adapters produce sharper images with clearer text edges, which is also supported by the higher PSNR metric. In short, compared to StbaleSignature, WMAdapter produces higher quality images with fewer noticeable artifacts. Appendices A.6, A.7, A.8 provide additional comparisons across more datasets.

## 5. Conclusion and Limitation

In this paper, we introduce WMAdapter, a plug-and-play watermarking plugin that enables latent diffusion models to embed arbitrary bit information during image generation. Our adapter is lightweight, easy to train, and offers a superior accuracy-quality trade-off with significantly fewer noticeable artifacts compared to previous post-hoc and diffusion-native watermarking methods. One limitation is that the Adapter-*F* variant occasionally produces grid-like artifacts that become visible upon zooming in. In summary, WMAdapter provides a simple yet powerful baseline for further exploration on diffusion watermarking.

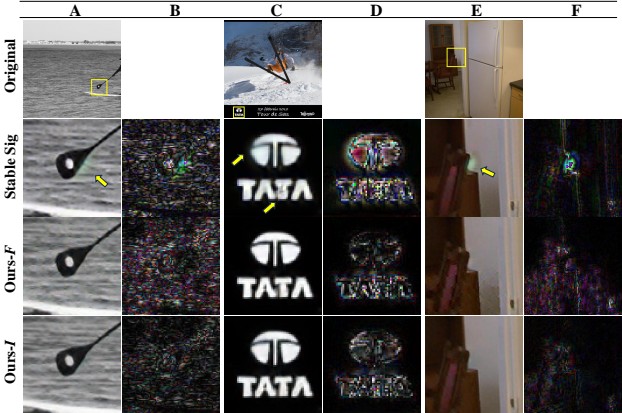

Figure 7: Comparison between WMAdapter and StableSignature (Fernandez et al., 2023). Yellow arrows point to the generated artifacts. *(B)(D)(F)* show the difference after watermarking. View in color and zoom in.

## Impact Statement

This paper presents work whose goal is to advance the field of Machine Learning. There are many potential societal consequences of our work, none which we feel must be specifically highlighted here.

## Acknowledgment

This project is supported by the Ministry of Education, Singapore, under the Academic Research Fund Tier 1 (FY2022) Award 22-5406-A0001.

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

# A. Appendix

## A.1. Experiment Statistical Significance

For the first training stage, we ran 3 independent training and found the standard deviation of average validation bit accuracy across 3 runs to be 0.0006, and the standard deviation of validation PSNR to be 0.03 dB.

For the second finetuning stage, we also ran 3 independent trials. The standard deviation of average validation bit accuracy across 3 runs was also 0.0006, and the std of validation PSNR was 0.04 db. The small standard deviation at both stages demonstrates the stability of our method. Since the standard deviation is too small to be clearly viewed in Fig. 8, we report the numbers in text.

## A.2. Broader Impacts

The proposed diffusion watermarking technique offers significant positive societal impacts, such as enhancing copyright protection for digital creators and helping to prevent the spread of fake news by enabling the authentication of images. However, it also poses potential negative impacts, including privacy concerns, the risk of misuse for malicious purposes, technical challenges that may disadvantage smaller creators, and possible degradation of image quality. Balancing these benefits and drawbacks is crucial to ensure the responsible and effective use of this technology.

In terms of applications, our proposed WMAdapter can also be directly applied to video generation models such as AnimateDiff (Guo et al., 2023) and StableVideoDiffusion (Blattmann et al., 2023), which share the same VAE architecture as image Diffusion models. We leave further exploration on video to the future work.

## A.3. Evaluation on Various Distortion Intensities

Fig. 8 evaluates our method under larger ranges of distortion intensities and more attacks. We can see that our adapters remain comparable robustness to Stable Signature (Fernandez et al., 2023) over range of attack intensities. Note that all three methods exhibit limited robustness to significant Gaussian noise and Rotation. This limitation arises because the pretrained HiDDeN decoder (Fernandez et al., 2023) was not specifically trained to handle such attacks. To further enhance robustness under such attacks, WMAdapter would need to be built upon a watermark decoder that is pretrained with rotation and noise augmentation.

## A.4. Visualization of Distortions

Fig. 9 shows different image distortions evaluated in the paper.

## A.5. Evaluation Against Other Adaptive Attacks

We also evaluate WMAdapter-*I* against another adversarial attack Lukas et al. (2023), which propose to train a stronger surrogate detector. We reproduce the adversarial noising method described in the paper. Specifically, we implemented their approach using the reported hyperparameters. We found that the suggested $\epsilon$-ball of 2/255 produced negligible attack effects. We increased the $\epsilon$-ball to 8/255, reducing PSNR from 34.8 to 30.3 (similar drop to other attacks in our Fig. 5), while the bit accuracy dropped moderately from 0.98 to 0.93. This suggests that our method demonstrates resilience to such attacks.

## A.6. More Qualitative Results on COCO Dataset

Fig. 10 shows the watermarked images and their difference with the original images. We find that both WMAdapter-*F* and WMAdapter-*I* can adaptively embed watermark information into regions with significant color variations and richer textures in the images, significantly enhancing their invisibility.

## A.7. Generalization to Ideogram Dataset

Fig. 11 shows our results on images generated by Ideogram (Ideogram.ai, 2024). These images exhibit completely different styles. However, our WMAdapter, trained on COCO, transfers seamlessly to them.

### A.8. Comparison With Other Watermarking Methods

Fig. 12 compares various watermarking methods. We observe that our method introduces minimal noticeable artifacts to the images. Thanks to the dedicated design of the contextual adapter, the modifications adapt more effectively to the cover image content.

While the JND enhancement (Fernandez et al., 2022a) used by HiDDeN* can also adapt the watermark post-hoc. However, such post-hoc methods compromises robustness and tends to alter the background. In contrast, our contextual adapter is trained end-to-end, offering a better robustness-quality tradeoff (see Tab. 2 and Fig. 1).

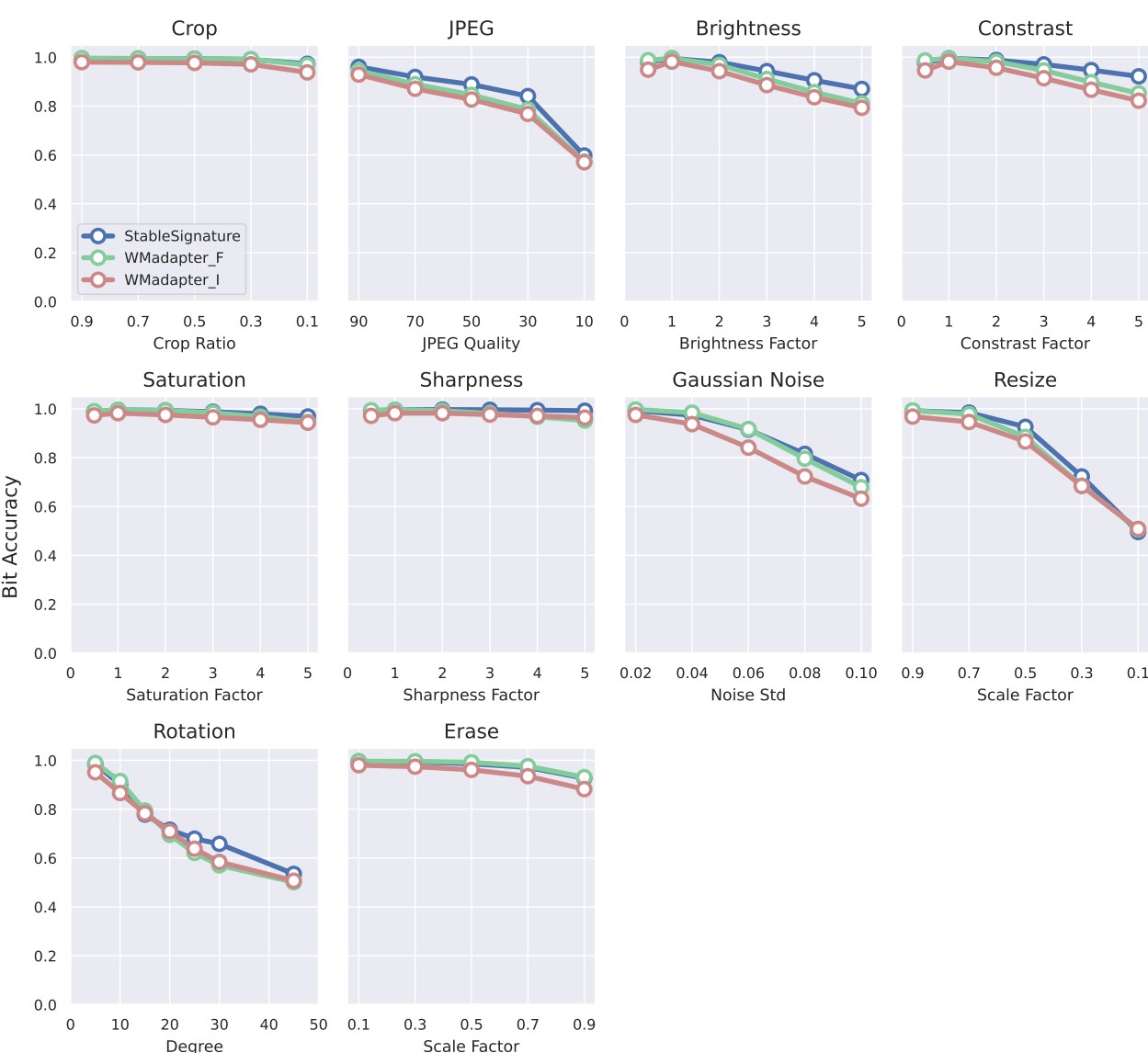

Figure 8: Robustness comparison over various distortion intensities.

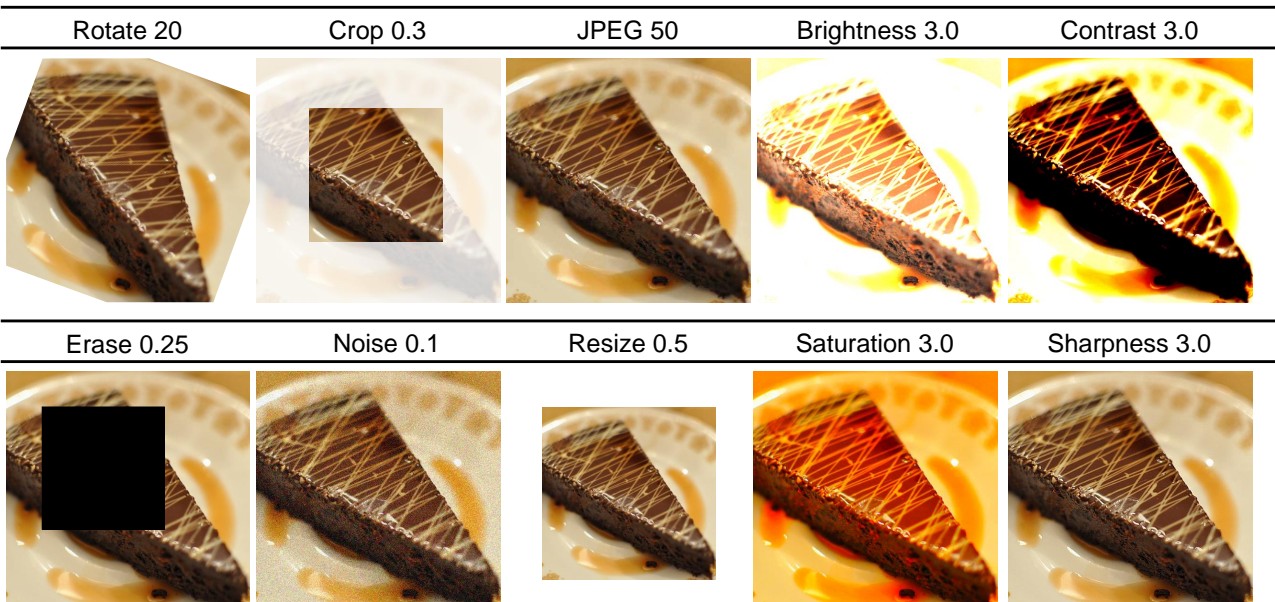

Figure 9: Visualization of different augmentations.

| Original | Adapter-*F* | Difference | Adapter-*I* | Difference |
|----------|-------------|------------|-------------|------------|

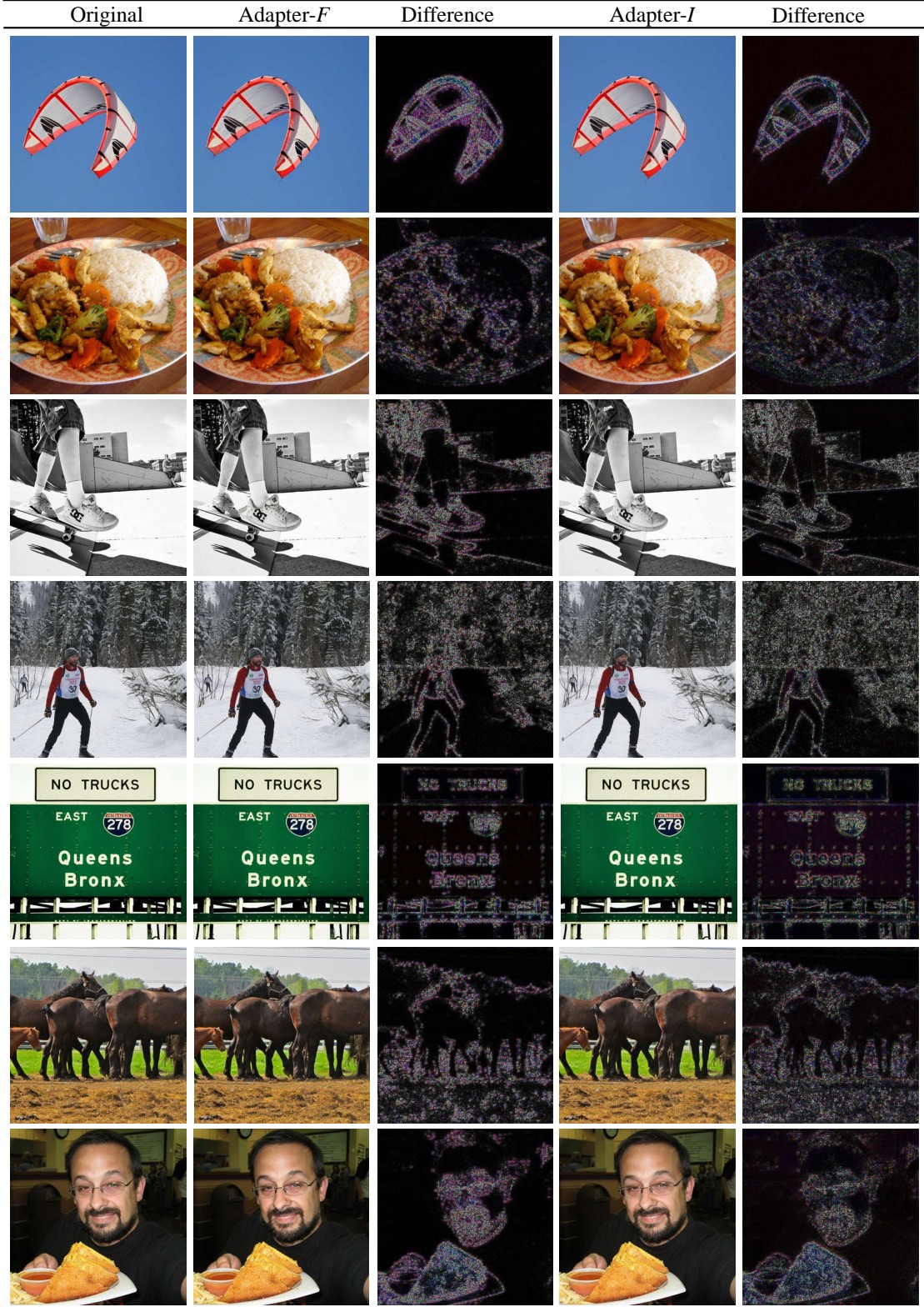

Figure 10: Qualitative results on COCO dataset at resolution 512.

| Original | Adapter-*F* | Difference | Adapter-*I* | Difference |
| --- | --- | --- | --- | --- |

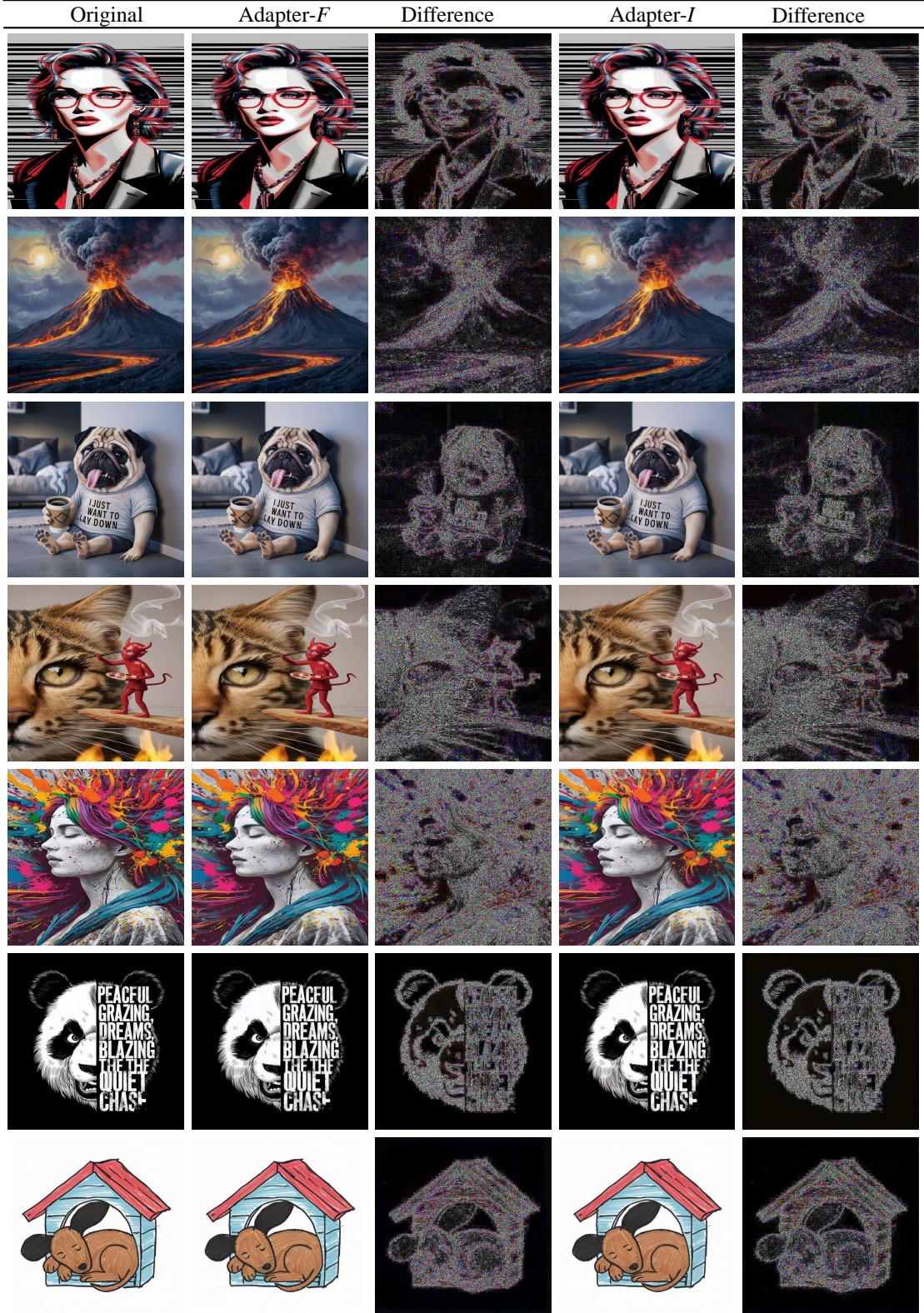

Figure 11: Qualitative results on Ideogram (Ideogram.ai, 2024) at resolution 512.

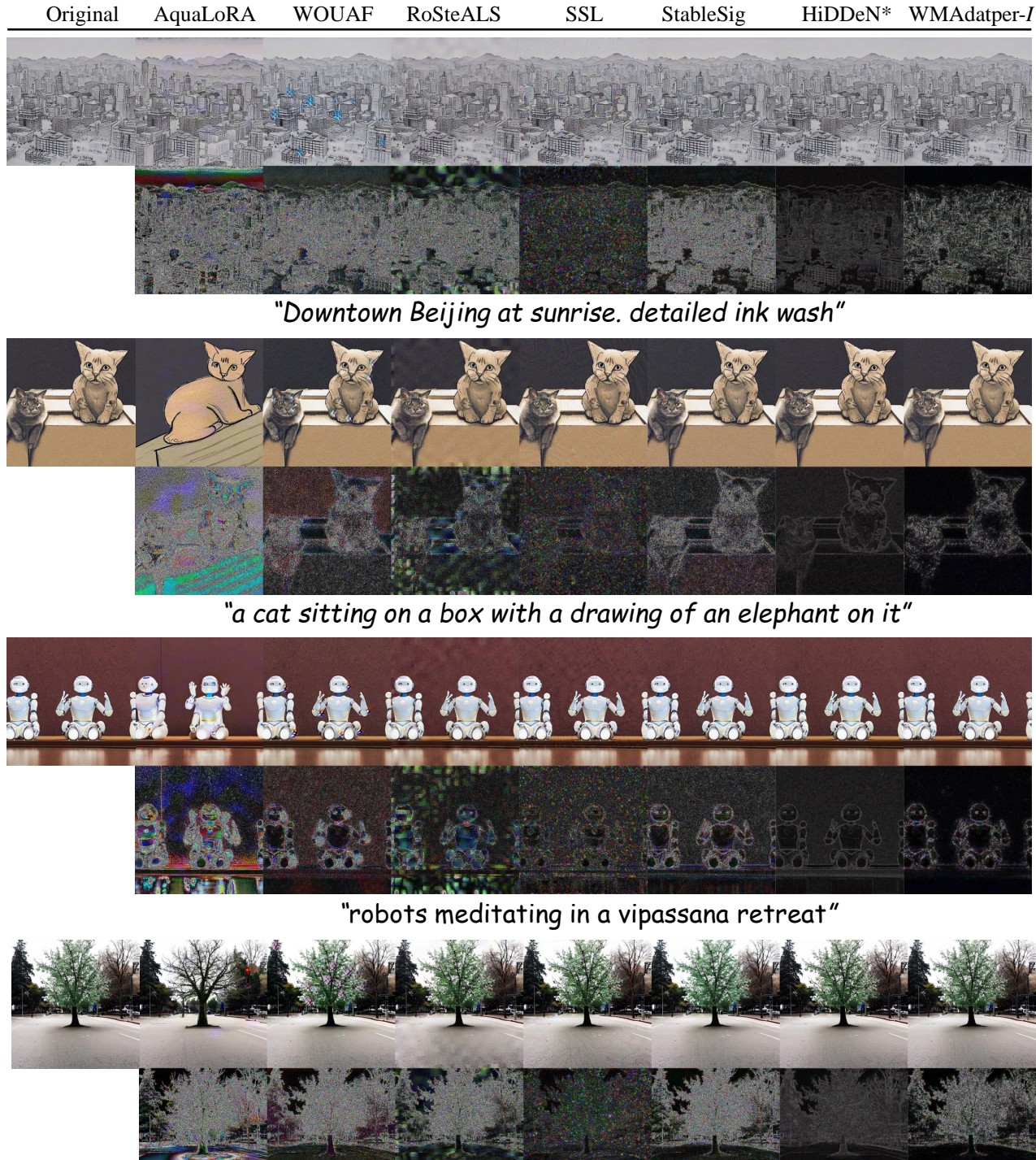

Figure 12: Watermarking images generated with given prompts. For HiDDeN* (Zhu et al., 2018), we use a post-hoc just noticeable difference (JND) mask to enhance invisibility (Fernandez et al., 2022a). Zoom in for best view.

