# OpenReview forum: "WMAdapter: Adding WaterMark Control to Latent Diffusion Models"
_ICML.cc/2025/Conference — ICML 2025 poster_

### Official Review · Reviewer_5xGq · 2025-03-11

**Overall Recommendation:** 3

**Summary:**

This paper introduces WMAdapter, a plug-and-play watermarking solution for latent diffusion models that embeds watermarks during the image generation process without modifying the original diffusion components. The authors propose two key innovations: (1) a contextual adapter that conditions on the content of the cover image to generate adaptive watermark embeddings, and (2) a hybrid finetuning strategy that preserves the integrity of diffusion components while suppressing artifacts.

**Claims And Evidence:**

The claim of superior image quality is backed by quantitative metrics (PSNR, FID) and qualitative visual comparisons showing fewer artifacts than other methods.

The comparison with AquaLoRA seems to focus on visual quality differences without acknowledging that AquaLoRA was designed for more stringent white-box attack scenarios.

**Essential References Not Discussed:**

Leveraging Optimization for Adaptive Attacks on Image Watermarks

**Experimental Designs Or Analyses:**

The experimental designs are generally sound.

**Methods And Evaluation Criteria:**

The watermarking architecture (using contextual adapters attached to VAE decoder) is well-motivated and explained clearly.

The hybrid finetuning strategy addresses a real issue (artifacts) observed in preliminary experiments.

However, the evaluation has some notable limitations:

- The paper acknowledges the quality-robustness trade-off but appears to prioritize visual quality over robustness in their design choices. This is a legitimate approach, but it would benefit from more explicit discussion, as robustness is generally considered more critical for watermarking applications.
- The paper lacks some important adaptive evaluations. In particular, one important recent works are not addressed: "Leveraging Optimization for Adaptive Attacks on Image Watermarks", These studies present important challenges to watermarking methods that should be evaluated.

**Other Comments Or Suggestions:**

* Why is there a discrepancy between the methods shown in the qualitative demonstration (left side of Figure 1) and the quantitative evaluation (right side of Figure 1)?
* Figure 8 demonstrates robustness against many additional distortions, but Table 2 only presents a limited subset. Why weren't all these distortions quantitatively evaluated in the main results?
* The paper lacks evaluation of rotation robustness, which is a critical transformation for watermark assessment. How does WMAdapter perform against rotation attacks, and why was this evaluation omitted?
* Recent works such as "Leveraging Optimization for Adaptive Attacks on Image Watermarks". How does WMAdapter resist these targeted attacks?
* The paper appears to prioritize visual quality over robustness. Given that robustness is generally considered more critical for watermarking applications, can you discuss this design choice and the explicit trade-offs made?
* How does the approach scale to larger or smaller watermark sizes beyond the 48-bit watermark used in experiments? Would it require retraining the entire system or just the adapter?
* Would the watermark still be recoverable after significant editing?
* Some minor typos, like in L400, "StbaleSignature" instead of "StableSignature"

**Other Strengths And Weaknesses:**

Strengths:

* The paper addresses a clear practical need for high-quality watermarking in diffusion models
* The solution is elegant, lightweight (1.3MB) and computationally efficient (30ms per image)
* The hybrid finetuning strategy is novel and effective at preserving image quality
* The approach is practical and implementable, with clear explanations of design choices

Weaknesses:

* The work prioritizes visual quality over robustness, which may limit its practical application in scenarios where security is paramount
* The evaluation lacks assessment against rotation attacks, which are common in real-world scenarios
* The paper does not evaluate against state-of-the-art adaptive removal methods using optimization or generative models
* There is inconsistency between the methods shown in qualitative and quantitative evaluations (Figure 1)
* Some attacks demonstrated in Figure 8 are not thoroughly quantified in the main results (Table 2)

**Questions For Authors:**

See above.

**Relation To Broader Scientific Literature:**

The paper effectively positions WMAdapter within the broader context of: 1) Traditional post-hoc watermarking methods (frequency domain, optimization-based, encoder-decoder) and 2) Diffusion-native watermarking approaches (initial noise-based vs. latent space-based).

**Theoretical Claims:**

The paper is primarily empirical rather than theoretical.

---

> ### Author Rebuttal · Authors · 2025-03-31
>
> **Q: Discrepancy between qualitative and quantitative evaluation in Figure 1.**
>
> Different types of watermarks introduce different types of artifacts. Although FID is one of the most widely used quantitative metrics for image quality, it often struggles to accurately reflect the visual impact of diverse artifacts. Bridging the gap between quantitative metrics and human perception remains an open research problem and an active area of investigation.
>
> &nbsp;
>
> **Q: Why weren't all distortions in Figure 8 quantitatively presented in Table 2**
>
> Due to space limitations, we had to make a trade-off in presentation. In Table 2, our goal was to provide a balanced overview of both image quality and robustness metrics. Including all attack types and quality metrics in a single table would have been impractical. This design choice is similar to that of Table 1 in *Stable Signature*. To complement Table 2, we included Figure 8, which provides a more comprehensive evaluation by illustrating robustness under a wider range of attack types and intensities.
>
> &nbsp;
>
> **Q: How does WMAdapter perform against rotation attacks?**
>
> Please refer to this anonymous [link](https://ibb.co/FjJWm3c) for our evaluation of rotation robustness. It is important to note that WMAdapter is built upon the public checkpoint provided by *Stable Signature*, which was not pretrained with rotation augmentation. Despite this, WMAdapter demonstrates robustness to moderate rotation (up to 15 degrees) and performs comparably to *Stable Signature*. To further enhance robustness under stronger rotation, WMAdapter would need to be built upon a watermark decoder that is pretrained with rotation-augmented data.
>
> &nbsp;
>
> **Q: “The paper does not evaluate against state-of-the-art adaptive removal methods using optimization or generative models”**
>
> We would like to clarify that our paper **does evaluate against SOTA opensource** adaptive removal methods, including both optimization-based and generative approaches. Specifically, for **optimization-based methods**, we include evaluations against the adversarial attack proposed in [An’24] under both black-box and white-box settings. For **generative models**, we assess performance against VAE-based methods ([Cheng’20], [Balle’18]) as well as a diffusion-based method ([Zhao’23]). Please refer to Sec. 4.3 and Fig. 5 for detailed results.
>
> &nbsp;
>
> **Q: Lack evaluation against "Leveraging Optimization for Adaptive Attacks on Image Watermarks”**
>
> As the code for the referenced paper is **not publicly available**, we have made our best effort to reproduce the adversarial noising method described in the paper. Specifically, we implemented their approach using the reported hyperparameters. We found that the suggested $\epsilon$-ball of 2/255 produced negligible attack effects. We increased the $\epsilon$-ball to 8/255, reducing PSNR from 34.8 to 30.3 (similar drop to other attacks in our Fig.5), while the bit accuracy dropped moderately from 0.98 to 0.93. This suggests that our method demonstrates resilience to such attacks. We'll include the result in updated version.
>
> &nbsp;
>
> **Q: Discuss the design choice and the explicit trade-offs made between visual quality and robustness”**
>
> Our goal is to develop a high-quality, artifact-free watermarking technique. To this end, we propose a **non-intrusive watermarking framework** that avoids modifying the pretrained diffusion pipeline. So we design **Contextual Adapter** as a plugin avoiding direct modification of diffusion models. Also we develop **Hybrid Finetuning** to conduct non-intrusive training.
>
> The relative importance of robustness versus visual quality is indeed a nuanced and context-dependent question. From the perspective of watermarking as a standalone technique, **robustness** is often considered the primary metric. However, when we consider broader deployment scenarios—such as integrating watermarking into GenAI products, which is a key motivation of our work—the priorities shift. In these contexts, **visual quality becomes paramount, as it directly impacts user experience and product adoption**. There is a strong practical demand for artifact-free watermarking that preserves the high visual fidelity expected from GenAI products. Therefore, our design values visual quality without compromising essential robustness, striking a balance suitable for real-world applications.
>
> &nbsp;
>
> **Q: “How does the approach scale to larger or smaller watermark sizes beyond the 48-bit watermark?”**
>
> Our approach is built on a pretrained watermark decoder. To scale to different watermark sizes, one can simply replace the decoder with a pretrained version of the desired size and retrain only the adapter—the rest of the system remains unchanged. Alternatively, users may train a watermark decoder of arbitrary size themselves.
>
> &nbsp;
>
> **Q: Robustness to significant editing**
>
> Thanks for your suggestion. We randomly edit 80% of the image by inpainting, the bit accuracy remains 0.91.

---

> > ### Comment · Reviewer_5xGq · 2025-04-08
> >
> > Thanks for the author's feedback. My concerns are mainly addressed. I tend to be positive!

---

### Official Review · Reviewer_BZ2P · 2025-03-12

**Overall Recommendation:** 3

**Summary:**

This paper introduces WMAdapter, a watermarking plugin for AI-generated images that seamlessly embeds user-specified watermark information during the diffusion generation process. Unlike previous methods that modify diffusion modules to embed watermarks, WMAdapter preserves the integrity of diffusion components, resulting in sharp images with no noticeable artifacts. The contributions of this paper include a contextual adapter for generating adaptive watermark embeddings based on the image content and a hybrid finetuning strategy to suppress artifacts while preserving the integrity of the diffusion process. Experimental results show that WMAdapter provides superior image quality, flexibility, and competitive watermark robustness.

**Claims And Evidence:**

The authors claim that WMAdapter is a novel diffusion watermarking solution with an innovative design philosophy, which is appropriate. They also claim that WMAdapter outperforms previous post-hoc and diffusion-native watermarking methods, and the experimental results support this claim.

**Essential References Not Discussed:**

[1] Yang Z, Zeng K, Chen K, et al. Gaussian shading: Provable performance-lossless image watermarking for diffusion models. Proceedings of the IEEE/CVF Conference on Computer Vision and Pattern Recognition. 2024: 12162-12171.
[2] Xiong C, Qin C, Feng G, et al. Flexible and secure watermarking for latent diffusion model. Proceedings of the 31st ACM International Conference on Multimedia. 2023: 1668-1676.

**Experimental Designs Or Analyses:**

The experimental and analyses of this paper is sound and valid. It compared some important model watermarking methods, such as Stable Signature.

**Methods And Evaluation Criteria:**

The proposed methods and evaluation criteria make sense for the diffusion model watermarking.

**Other Comments Or Suggestions:**

Please refer to the weakness.

**Other Strengths And Weaknesses:**

**Strength**

(1) The paper proposes a watermarking method for image generative models that demonstrates robustness against various types of attacks, outperforming existing approaches.

(2) It explores a hybrid training strategy involving the VAE decoder, which effectively mitigates the degradation of image quality caused by watermark embedding.

(3) Comprehensive ablation studies are conducted to evaluate the contribution of the adapter and hybrid training strategy, showing (i) improved watermark decoding accuracy and (ii) no compromise in the performance of the generative model.

**Weakness**

(1) It is recommended to include a comparison of watermark effects and potential artifacts under both hybrid training and standalone training modes.

(2) The motivation lacks clarity — how do the observations from existing methods logically lead to the need for fine-tuning the VAE decoder and the inclusion of associated modules?

**Questions For Authors:**

None

**Relation To Broader Scientific Literature:**

None

**Theoretical Claims:**

This paper does not provide any theoretical claims.

---

> ### Author Rebuttal · Authors · 2025-03-31
>
> **Q: Essential References Not Discussed: Gaussian Shading and FSW**
>
> Thank you for the suggestion. In fact, we have already referenced and discussed both works in the Introduction section. Specifically, they are cited as [Yang et al., 2024] and [Xiong et al., 2023], corresponding to Gaussian Shading and FSW, respectively.
>
> &nbsp;
>
> **Q:  “It is recommended to include a comparison of watermark effects and potential artifacts under both hybrid training and standalone training modes.”**
>
> Thank you for the suggestion. We kindly refer the reviewer to Figure 6, which provides a visual comparison of watermark effects and potential artifacts under different training modes. We will include more examples in the revised manuscript to improve clarity.
>
> &nbsp;
>
> **Q:  “The motivation lacks clarity — how do the observations from existing methods logically lead to the need for fine-tuning the VAE decoder and the inclusion of associated modules?”**
>
> We would like to further clarify our motivation. Existing methods typically watermark diffusion models by modifying the backbone or VAE decoder parameters, which often results in visible artifacts and degraded visual quality. To address this, we propose a **non-intrusive framework** that preserves all original diffusion parameters. Specifically, we design a **Contextual Adapter** that is attached to the VAE decoder, avoiding direct modification of its internal structure. In addition, we introduce a **Hybrid Finetuning Strategy** to train the Contextual Adapter in a similarly non-intrusive fashion.
>
> If there are any remaining questions regarding our motivation, we would be happy to provide further clarification.

---

### Official Review · Reviewer_ZSgp · 2025-03-12

**Overall Recommendation:** 4

**Summary:**

This paper proposes the **WMAdapter**, which generates content-aware watermark embeddings using the contextual adapter and embeds watermarks with a hybrid fine-tuning strategy. Specifically, the contextual adapter comprises a series of fuser modules, each of which is attached before a corresponding VAE decoder block. In addition to training the adapter, **WMAdapter** adopts a hybrid fine-tuning strategy that jointly optimizes the adapter and VAE decoder while using the original VAE decoder for inference. Experimental results show that the **WMAdapter** retains effective watermark-embedding capability while maintaining generation quality.

### Update after rebuttal
I've no further questions regarding this paper after rebuttal and keep my original score.

**Claims And Evidence:**

The claims are well supported.

**Essential References Not Discussed:**

No.

**Experimental Designs Or Analyses:**

The experiments in this paper seem sufficient.

**Methods And Evaluation Criteria:**

The proposed methods and evaluation criteria make sense. Below, I list one concern about hybrid fine-tuning.

1. I'm confused about the design of hybrid fine-tuning, and my concern is explained as follows. The better generation performance is achieved at the cost of detection capability. When jointly fine-tuning the adapter $\mathcal{A}$ and the VAE decoder $\mathcal{V}^{\textrm{new}}$, after several updating steps we have the loss $\mathcal{L}(\mathcal{A}(w), \mathcal{V}^{\textrm{new}}(f))$ which should be lower than $\mathcal{L}(\mathcal{A}(w), \mathcal{V}^{\textrm{ori}}(f))$, where $\mathcal{V}^{\textrm{ori}}$ indicates the original VAE decoder. This aligns with the experimental results where Adapter-I compromises parts of the watermarking capability to enhance the image quality. If so, my concern is whether it is possible to select a set of better $\lambda$ to achieve this without hybrid fine-tuning or find better checkpoints during the training process.

**Other Comments Or Suggestions:**

Please see above.

**Other Strengths And Weaknesses:**

The overall paper is well-written, and the method design is well supported by the motivation and experiments.

**Questions For Authors:**

Please see above.

**Relation To Broader Scientific Literature:**

WMAdapter can improve traceability within LDMs and reduce the misuse of AIGC contents [1].

[1] Security and Privacy on Generative Data in AIGC: A Survey

**Theoretical Claims:**

No theoretical proofs to check.

---

> ### Author Rebuttal · Authors · 2025-03-31
>
> **Q: Is it possible to select a better $\lambda$ or or find better checkpoints to enhance image quality?**
>
> Thank you for your insightful question. In practice, selecting a better $\lambda$ or checkpoint to enhance image quality proves very challenging. During our joint training experiments (Adapter-V), we extensively explored different $\lambda$ values and checkpoints. However, none could achieve a better trade-off between image quality and robustness compared to Hybrid Finetuning. Specifically, increasing the weight of the image quality loss significantly deteriorated bit accuracy, sometimes even leading to training collapse. Moreover, merely raising the quality loss weight does not effectively suppress the lens flare artifacts shown in Figure 6; it only slightly improves the quantitative metrics.
>
> We believe this issue arises not only from the choice of loss weights or checkpoints but also from modifications to the VAE decoder itself. Joint finetuning significantly differs from the large-scale pretraining originally conducted for the VAE. It adjusts full VAE decoder parameters based on much smaller datasets with distinct distributions, batch sizes, and training objectives (bit acc + quality). Consequently, this easily disrupts the inherent knowledge encoded in the pretrained VAE decoder, leading to reduced image quality or the emergence of artifacts.
>
> In other words, adjusting the loss weight or selecting different checkpoints during joint finetuning cannot effectively resolve the substantial deviations from the pretrained parameter landscape. The most reliable solution remains using the original pretrained VAE decoder directly, as employed in our proposed Hybrid Finetuning method.
>
> Regarding the loss, lower values of $\mathcal{L}(\mathcal{A}(w), \mathcal{V}^{new})$ achieved during joint fine-tuning do not necessarily indicate genuine improvements, as the VAE decoder $\mathcal{V}^{new}$ tends to overfit on the limited training data.
>
> &nbsp;
>
> **Q: Relation To Broader Scientific Literature**
>
> Thank you for highlighting this point. We will clarify and discuss the relation of our work to the broader scientific literature in the revised manuscript.

---

### Decision · Program_Chairs · 2025-05-01

**Decision:**

Accept (poster)

**Comment:**

The paper proposes a novel watermarking technique for diffusion models, more concretely an adapter (the WMAdapter) allowing to keep the components of the diffusion model intact and a fine-tuning strategy for training the adapter equipped diffusion model. Experimental results show that the WMAdapter retains effective watermark-embedding capability while maintaining generation quality.

The rebuttal clarified the open questions of the two reviewers and all voted for acceptance.